# Collaborative Autonomy: Human–Robot Interaction to the Test of Intelligent Help

**Filippo Cantucci * and Rino Falcone ***

Institute of Cognitive Science and Technology, National Research Council of Italy (ISTC-CNR), 00185 Rome, Italy
* Correspondence: filippo.cantucci@istc.cnr.it (F.C.); rino.falcone@istc.cnr.it (R.F.)

**Abstract:** A big challenge in human–robot interaction (HRI) is the design of autonomous robots that collaborate effectively with humans, exposing behaviors similar to those exhibited by humans when they interact with each other. Indeed, robots are part of daily life in multiple environments (i.e., cultural heritage sites, hospitals, offices, touristic scenarios and so on). In these contexts, robots have to coexist and interact with a wide spectrum of users not necessarily able or willing to adapt their interaction level to the kind requested by a machine: the users need to deal with artificial systems whose behaviors must be adapted as much as possible to the goals/needs of the users themselves, or more in general, to their mental states (beliefs, goals, plans and so on). In this paper, we introduce a cognitive architecture for adaptive and transparent human–robot interaction. The architecture allows a social robot to dynamically adjust its level of collaborative autonomy by restricting or expanding a delegated task on the basis of several context factors such as the mental states attributed to the human users involved in the interaction. This collaboration has to be based on different cognitive capabilities of the robot, i.e., the ability to build a user's profile, to have a Theory of Mind of the user in terms of mental states attribution, to build a complex model of the context, intended both as a set of physical constraints and constraints due to the presence of other agents, with their own mental states. Based on the defined cognitive architecture and on the model of task delegation theorized by Castelfranchi and Falcone, the robot's behavior is explainable by considering the abilities to attribute specific mental states to the user, the context in which it operates and its attitudes in adapting the level of autonomy to the user's mental states and the context itself. The architecture has been implemented by exploiting the well known agent-oriented programming framework Jason. We provide the results of an HRI pilot study in which we recruited 26 real participants that have interacted with the humanoid robot Nao, widely used in HRI scenarios. The robot played the role of a museum assistant with the main goal to provide the user the most suitable museum exhibition to visit.

**Keywords:** human–robot interaction; user modelling and user adaptation; theory of mind; social adjustable autonomy; explainabile agency

## 1. Introduction

This work addresses the problem of how to build autonomous robots that are able to effectively cooperate with humans, with capabilities similar to those exhibited by humans when they cooperate with each other. Even with sufficiently safe and predictable behaviors, robots will never be *socially* adequate until they behave with the abilities shown in human–human cooperation [1–3]. Robots are part of daily life and are present in multiple environments such as cultural heritage sites [4,5], hospitals [6], tourist scenarios [7] and so on. In these contexts, robots interact with a wide spectrum of users not necessarily able (or willing) to adapt their interaction level to the kind requested by a machine: the users need to deal with artificial systems whose behaviors must be understandable [8] and personalized [9]. Let us consider the interactive scenario involving a human user and a robot: the human relies on the robot for realizing some part of her plan. Now, in order to help the human in an intelligent, adaptive and effective way, the robot has to understand

the human's goals, beliefs and expectations about its behavior. Furthermore, intelligent help must be much more than simple execution of a prescribed action; it requires autonomy and even initiative. In particular, *collaborative autonomy*: an autonomy at the service of the user. Generally speaking, the robot should provide to the user different levels of help [10]: it should exploit its autonomy, competence and cognitive skills to find a better or a possible solution for the user's goals, that does not necessarily match with her explicit expectations. This should not necessarily require a discussion or an agreement: it might be an independent initiative of the robot. How would this advanced form of true cooperation and partnership be possible without a reciprocal mind ascription and reading? A robot needs to understand the human's ends, higher goals, expectations interests in order to really help her, and the human has to understand what the robot is doing or did and why in mental terms. What was the knowledge or the beliefs of the system, its helping aim, its reasoning and problem solving? For being socially effective, autonomous robots have to build or use cognitive models of the other agents [11] and to adapt their decisions, including the *level of autonomy* in achieving tasks, on the basis of the goals/needs of their interlocutors.

Contextual to the ability to consider the user needs and build complex models of her mental states, an intelligent robot should also take into account the interests, goals and plans of other agents involved in the interaction. These plans, goals and interests typically represent implicit restrictions and mandatory choices that the user has to consider; however, they can be adapted and *personalized* for each user. Based on the mental attitudes attributed to the user, and the constraints or needs attributable to other agents involved in the interaction, a mediation system between these two subjects can play a role of customization in order to best satisfy both parties. For example, we consider a scenario in which a robot plays the role of a museum assistant and has to personalize a museum visit according to the goals and interests of the user who intends to visit the museum. In this context, the robot not only personalizes a visit based on the user's artistic interests and other characteristics declared or attributable to her (e.g., time available, level of interests and so on), but it should also consider all those features related to the interests, goals and plans that the museum curators designed for a museum tour. Most of the time (this is the approach followed in this paper), the goals and interests of the museum curators are oriented to the satisfaction of the user, not in contrast to it (e.g., intent to guide the user to visit a really relevant collection that the user did not know and cannot assess the value of). However a negotiation process is necessary; in our case, this process takes place through the role of the mediator (e.g., a robot). In any case, it will be the user at the end of the visit who declares satisfaction about the mediation process realized by the robot. The type of collaboration mentioned above, which takes care of mental states and interests not declared by the user in addition to those explicitly requested or investigated by the robot, can be considered misleading or incorrect by the user. For this reason, a robot that intends to follow such a task adoption strategy must also transparently show the coherence of its behavior. A way to do that is to make its actions and goals *explainable* so as not to be confused with disordered and random help [12]. A robot—in this case—has to explain to the user what it did or plans to do in terms of the *reasons* for doing so.

*Contributions*

In this paper, we extend our previous work [13], and we introduce a knowledge-driven, reasoning-based, cognitive architecture for a robot's decision making whose result is an *intentional system* [14] able to:

- Build a profile of the interacting user and classify her on the basis of the robot's capability to perceive, infer or explicitly investigate specific physical and social features of the user;
- Model the mental states of the interacting human user in terms of beliefs, goals and plans and create a complex user model;
- Model the beliefs, goals and plans of other agents involved in the interaction;

- Dynamically adjust its own level of collaborative autonomy by restricting or expanding a delegated task on the basis of an internal negotiation process between several context factors, such as needs or goals not necessarily declared by the interacting human user and the constraints or needs attributable to other agents involved in the interaction; the final result provided by the robot does not necessarily correspond to the explicit and declared user request, and it is adapted to those mental states.
- Generate an explanation of the robot adoption strategy and justify the possible changes with respect to the explicit agreements.
- Investigate different dimensions of the user's satisfaction about the final results provided to the user.

We provide results obtained in a human–robot interaction pilot study with 26 real participants that have interacted with the humanoid robot *Nao* [15], widely used in HRI scenarios. The robot is a museum assistant that handles a virtual museum; its goal is to support the user and provide her a museum exhibition to visit. At the end of the tour, the robot proposes a short survey to the user, with the aim of investigating her satisfaction with respect to the presented exhibition. Furthermore, the robot explains the reasons that lead its task adoption process; after the explanation, it gives the user the possibility to change the answers to the administered survey. The computational model has been implemented by exploiting the well-known agent-oriented programming (AOP) framework *Jason* [16].

The paper is organized as follows: Section 2 describes the background underlying the cognitive architecture. Section 3 focuses on the description of the cognitive model; Sections 4 and 5 are dedicated to the pilot study and its results. Section 6 is dedicated to the conclusions and the future works. Finally, Section 7 introduces the state of the art related to the domain investigated in the pilot study: Cultural Heritage.

## 2. Background

In this section, we summarize the theoretical background on which the architecture design is based.

### 2.1. Cognitive Architectures

Cognitive architectures have been the subject of research for a long time, and different solutions have been proposed in order to provide artificial systems with human cognitive and behavioral characteristics [17–19]. Different approaches have been exploited, ranging from symbolic [20–22], emergent [23,24] and hybrid architectures [25,26]. The increasing utilization of robots in society determined a need to design architectures that also provide support for a broad range of HRI applications, where robots must share the same physical space [27], and they have to adapt the interaction to the users' needs, mental states and profiles [28–30]. Our architecture relates to the *Beliefs, Desires, Intentions* (BDI) paradigm [31], one of the most popular models in agent theory [32]. Originally inspired by the theory of *human practical reasoning* developed by Michael Bratman [33], the BDI model focuses on the role of intentions in reasoning and allows us to characterize agents using a human-like point of view. Very briefly, in the BDI model, the agent has *beliefs*, information representing what it perceives in the environment and communicates with other agents, and *desires* that are the states of the world that the agent would like to accomplish. The agent deliberates on its desires and decides to commit to some of them: committed desires become *intentions*. To satisfy its intentions, it executes *plans* in the form of a course of actions or sub-goals to achieve. The behaviour of the agent is thus described or predicted by what it committed to carry out. An important feature of BDI agents is the property to react to changes in their environment as soon as possible while keeping their pro-active behaviour. Different cognitive architectures underlie BDI modelling [30,34,35]. In this work, we extend upon the procedural reasoning system architecture described in [36], a framework for constructing real-time BDI reasoning systems that can perform complex tasks in dynamic environments.

### 2.2. Autonomy Adaptation in HRI

Autonomous robots are intelligent systems capable of performing tasks in the world without explicit human control. *Autonomy* in robotics has been widely defined, conceptualized and experimented with and is particularly important within the field of HRI, where it is not an end in itself, but rather a means for supporting productive interaction (active collaboration, assistance). Existing frameworks [37,38] provide definitions of autonomy and conceptualization of robot levels of autonomy in HRI scenarios. In contexts where humans and robots collaborate and share complex, dynamic spaces and tasks, determining the appropriate level of autonomy in a robot depends not only on "What can a robot do"? but rather on "What should a robot do and to what extent"? [38]. Selecting the appropriate level of autonomy and adapting it to the context involves both user control and robot initiative. Different approaches to autonomy and collaboration have been proposed, providing different balance levels between human control and autonomous decisions of the robot [39–41]. Our approach is based on the theory of adjustable social autonomy [42]; in the next section, we summarize the principles underlying the proposed cognitive architecture, focusing on the complex mental attitudes of task delegation and task adoption.

### 2.3. Theory of Mind

Everyday life scenarios increasingly involve sophisticated autonomous robots which cooperate with humans (mainly non-specialists) in complex environments. This *human-in-the-loop* role for humans interacting with intelligent robots requires that the latter are able to complement existing human capabilities or, even better, to autonomously *read* [43] the mental states of the humans and adapt their behavior with respect to their beliefs, goals, plans and features. The complex capability to represent mental states of other humans is broadly referred as *Theory of Mind* (ToM) [44,45]. The ability to mentalize other agents implies building models of them and using such models for different reasons: to predict the actions of the modelled agent and integrate the prediction in the decision-making process in order to improve the effectiveness of the interaction; to classify the behaviour of the modelled agents and choose a pre-designed strategy which is known to work in favour of the identified class and to exploit the model of a specific agent in order to identify and point out some characteristics. Providing intelligent robots with the ability to build complex models of their users and to modulate their decision on the basis of these models [11,46–48], represents a crucial point for promoting an intelligent and much more adaptive cooperation.

### 2.4. Theory of task Delegation and Adoption

As widely argued in [10], "*cooperation* works through the allocation of some task (or sub-task) by a given agent (individual or complex) to another agent [...], meeting some commitment"; in other words, cooperation implies the two complementary basic ingredients of task *delegation* and *adoption*. This kind of relationship is basic for any collaborative scenario where intelligent agents (human or not) are involved [49,50]. In cooperative scenarios, robots are not only useful for relieving humans of repetitive and boring tasks that they delegate to these artificial systems. To be really helpful, robots must take the initiative [51] of going beyond a delegated task or opposing human's expectation and, for example, proposing a different strategy for achieving the delegated goal [52].

Delegation and adoption can be represented by the following five-part relational constructs [10]:

$$Delegates(X, Y, C, \tau, g_X) \quad , \quad Adopts(Y, X, C, \tau, g_X) \qquad (1)$$

With the term *goal*, we identify a state of the world to be achieved. The first construct can be read as $X$—the *delegator*—delegates to $Y$—the *delegee*—to execute a task $\tau$, in a context $C$ and to realize the result $p$ (state of world) that includes or corresponds to $X$'s goal $Goal_x(g) = g_X$. The second construct can be read as $Y$—the *adopter*—adopts for/by $X$—the *adoptee*—to execute the task $\tau$ in context $C$ in order to realize the result $p$ (state

of world) that includes or corresponds to $X$'s goal $Goal_x(g) = g_X$. In simple cooperative scenarios, each delegation should correspond to the relative adoption and vice versa. The task $\tau$, the object of delegation/adoption, can be referred to as both an action $\alpha$ and its result $p$. Sometimes, either the action or the resulting world state is not explicitly stated, but, in any case, both will be present in at least implicit form. So by means $\tau$, we will refer to the (action/state of world) pair $\tau = (\alpha, p)$. For a complete formalization of the theory of delegation please refer to [10,42].

Let us focus on a deep level of cooperation where the delegee/adopter can adopt a task delegated by the delegator/adoptee, at different levels of effective help. The different levels of the delegee's (adopter's) adoption can be individuated according to [10]:

- *Sub-help* : The delegee/adopter satisfies a sub-part of the delegated world-state (so satisfying just a sub-goal of the delegator/adoptee),
- *Literal help*: the delegee/adopter adopts exactly what has been delegated by the delegator/adoptee,
- *Over help*: the delegee/adopter goes beyond what has been delegated by the delegator/adoptee without changing the delegator's (adoptee's) plan (but including it within a hierarchically superior plan),
- *Critical Over-help*: the delegee/adopter realizes an Over-help and in addition also modifies the original plan/action (included in the new meta-plan),
- *Critical help*: the delegee/adopter satisfies the relevant results of the requested plan/ action (the goal), but modifies that plan/action,
- *Critical Sub-help*: the delegee/adopter realizes a sub-help and in addition modifies the (sub) plan/action.

In our scenario, the role of delegator is attributed to the human user $H$, while the adopter is the robot $R$; $\alpha$ can be either a complex action (plan) or an elementary action. It is important to underline that we are considering collaborative robots whose main meta-goal is the achievement of the user's goals.

In the literature, the semantic distinction of the terms *cooperation* and *collaboration* is presented in various forms that are not always completely homogeneous, and the meanings are not always clearly distinguishable from each other. In fact, it can be argued that "the key difference between these approaches to group work is that cooperation is more focused on working together to create an end product, while successful collaboration requires participants to share in the process of knowledge creation" [53]. In practice, collaboration is the result of a "continued attempt to construct and maintain a shared conception of a problem" [54], i.e., a "mutual engagement of participants in a coordinated effort to solve the problem together" [54]. Cooperation, on the other hand, highlights a differentiation and separation of the activities of the participants: a sort of division of labor with different responsibilities on different portions of problem solving. Cooperation might be just functional, self-organizing and emergent, not necessarily mentally shared by the participants and intended, while collaboration implies some mental sharing and intentionality. In this paper, we found ourselves in a situation that is not entirely defined with respect to this differentiation and rather mixed. In fact, if it is true that it could be established that the human delegates a task to the robot and that the latter carries it out on its own (cooperative case), it is also true that the robot continuously needs to evaluate the environmental conditions in which it operates and at the same time verify how its activity is evaluated by the human in order to modify and better articulate its help in a key for adaptation to other goals and interests of the same human (collaborative case). For this reason we have used both terms in the course of this paper.

*2.5. Explainable Agency*

As mentioned in Section 1, the complex capability to ascribe a mind to other agents and behave according to their mental states is crucial in human-human interaction. In HRI scenarios, this ability is investigated [8], but limited for now. In order to reduce the impact of this lack, is responsibility of the robots designers to make them as transparent and

interpretable as possible: this requires the construction of representations that support the articulation of explanations [55]. Robots have to be able to explain their decisions [56] about the task they accomplish, mostly when they have an autonomous behaviour with respect to the task delegated by humans [57]. The definition *Explainable Agency* [58] refers to the capability of autonomous agents (e.g., robots) to explain their actions and the reasons leading to their decisions. Explainable agency stands out from *Explainable in Data-Driven domain* [59] which is referred to the capability to interpret the results of "black box" machine learning mechanisms such as deep learning. In this work we decide to leverage on a BDI based approach, in order to provide expressive explanations for goal-directed agents [60–62]. BDI paradigm is based on folk-psychology and this makes it suitable for providing human-friendly explanations, including conflicts justification. Indeed, BDI model allows explicit representations of the robot's mental states (beliefs, goals, plans and so on) and those ones the robot attributes to the interacting human agents; this allows to select beliefs, goals, plans, intentions, constraints and so on, that have been relevant in the entire deliberation process and exploit them in order to generate explanations about behavior/decisions, easily interpreted by humans. Furthermore, we will show how much explainability is important when conflicts arise between the results expected by the delegator and the ones provided by the adopter.

## 3. Cognitive Architecture Description

The proposed cognitive architecture defines a task-oriented robot that performs effective reasoning in a dynamic interaction with other cognitive agents, typically humans. The result is a cognitive robot able to receive a delegated task and to exert some degree of discretion, which corresponds to different levels of autonomy in the task adoption. We refer to social autonomy in a collaborative relationship among agents as the possibility of expanding the decision and action space with respect to the assigned task on the basis of criteria related to the adaptability to the user needs. This is possible thanks to the robot's capability to have a ToM of the user, even of mental states beyond those explicitly declared. In addition to adapting the task adoption strategy to the user's mental states, the robot is able to elicit an internal negotiation process in which it tries to mediate the final decision by taking into account mental state attributes of other agents involved in the interaction. Figure 1 shows the conceptual components of our cognitive architecture. In the next subsections, we will describe the main modules, following the perception–reasoning–action (PRA) cycle.

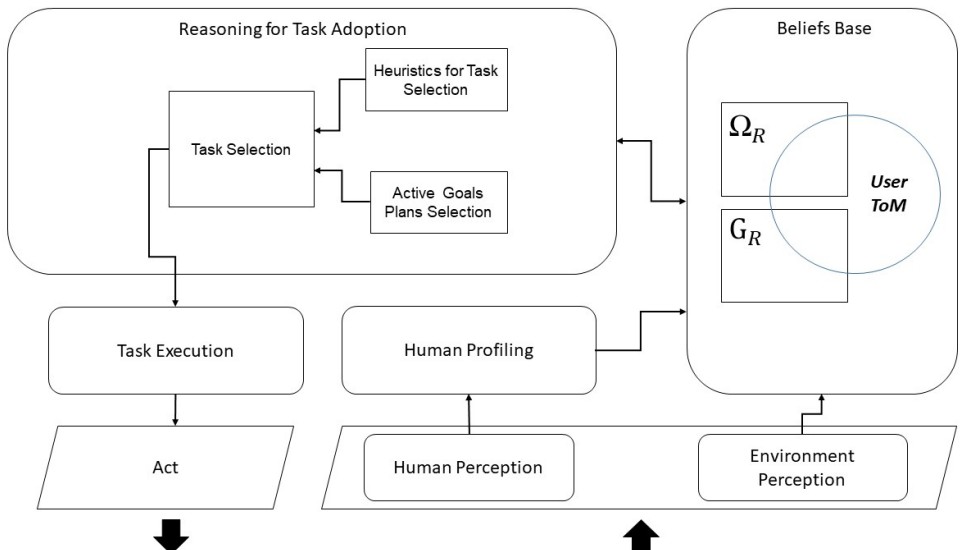

**Figure 1.** Perception–reasoning–action (PRA) cycle of the cognitive architecture. Please note that every module can manipulate (read or write) $G_R$ and $\Omega_R$.

### 3.1. The Robot Mental States

Mental states of the robot are stored in the *Robot's Beliefs Base* $\Omega_R$ and the *Robot's Goals Base* $G_R$ represented in the following subsets:

- $B_H = B_{sp_H} \cup B_{ms_H}$, the robot's beliefs about attributions of the human user $H$. In particular, $B_{sp_H} = \{\beta_{sp_{H1}}, \dots, \beta_{sp_{Hv}}\}$ is a subset of features, both social (i.e., user's economic status, level of education, personal interests) and physical (i.e., user's age, gender), the robot attributes to the user; $B_{ms_H} = \{\beta_{ms_{H1}}, \dots, \beta_{ms_{Hw}}\}$ are the user's the mental states defined on the basis of the robot's capability to have a ToM of her;

- $Bms_A = \{Bms_{A_1} \dots Bms_{A_n}\}$ are the robot's beliefs about attributions of the agents $\{A_1, \dots, A_n\}$ involved in the interaction. If we refer to the $i$-th agent $A_i$, we will represent her mental state as the subset $B_{ms_{A_i}} = \{\beta_{ms_{A_{i1}}}, \dots, \beta_{ms_{A_{iw}}}\}$. Unlike the user model, which is created at run time based on her profile, the model of the other agents involved in the interaction has been previously described in the robot's belief base. While representing an a priori knowledge of the robot, this model can be modified by the robot itself through interaction with the agents involved;

- $B_{I_R} = \{\beta_{I_{R1}}, \dots, \beta_{I_{Rm}}\}$ are the robot's beliefs about its own physical internal state (i.e., the robot's battery, the temperature of its joints and so on);

- $B_\Pi = \{\beta_{\pi_1}, \dots, \beta_{\pi_s}\}$ is a plan library representing the beliefs about the set of actions (either *elementary* or *complex*) able to achieve world states;

- $K_R = \{\beta_{k_{R1}}, \dots, \beta_{k_{Rh}}\}$ represent the rest of the robot's knowledge.

Let us specify that $\Omega_R = B_H \cup B_{I_R} \cup B_\Pi \cup K_R$, while $G_R = \varnothing$ is initially an empty set. This is because in our case, the robot's goals are included in $G_R$ during the overall task delegation and adoption process. The plan library that plays an important role in the robot's decision-making process is defined as follows:

$$\Pi = \Pi^a \cup \Pi^d \tag{2}$$

where $\Pi^a$ indicated abstract plans (i.e., walk, drive, run are a type of move), while $\Pi^d$ is the set of plans that can be decomposed in sub-plans (i.e., travel consists of leaving, moving and arriving). We consider the plan library as unique and common to all agents, both robots and humans. Each plan is a tuple $\pi = <p, \gamma, \lambda, b>$ (with $\pi \in \Pi$) that has a body $b$ (course of actions $\alpha_i$ to execute or sub-goals $g_i$ to achieve), preconditions $\gamma$, constraints $\lambda$ and states of the world $p$ it will realize if correctly applied. Please refer to [63,64] for a much more complete meaning of plans. The presence of the plan library makes the approach deterministic (actions are mapped in the plan library) and fully observable (the plan library is known).

### 3.2. Perceiving the World

The first thing the robot does is to continuously sense and monitor the environment. The *Perception* modules map the current state of the world (including the interacting human user), provided by multiple robot's perception capabilities (i.e., object detection, speech recognition, face recognition and so on) in a symbolic form. Every time a new user wants to interact, the robot starts a dialogue with her during which it retrieves information (explicit or implicit), including the task the user intends to delegate. The interaction can also be supported by graphical user interfaces (GUI) through which the user can express specific social features (e.g., personal interests in art, sport and so on), and the robot can collect useful data to profile her.

### 3.3. The Reasoning Process

The reasoning process for task adoption exploits the information gathered in $\Omega_R$ and $G_R$ in order to deliberate upon the state of affairs it wants to pursue (*deliberation activity*) and decides how to use the available plans in $\Pi$ for achieving the deliberated task (*means-end reasoning*).

### 3.3.1. User Modelling

The *Human Profiling* module allows the robot to map the outputs of its own perception in two tuples of beliefs $\mathcal{P}_H = <\beta_{sp_{Hk}}, \dots, \beta_{sp_{Hk+n}}> \in B_{sp_H}$ and $\tau_0 = <\alpha_0, p_0>$. $\mathcal{P}_H$ represents the *user's socio-physical profile* inferred by the robot on the basis of its capability to perceive and infer specific user's features, both physical and social (i.e., age, level of education, personal interests and so on); $\tau_0$ is the *task* the user intends to delegate to the robot. The task $\tau_0$ can be not only explicitly delegated to the robot, but it can also be derived by the robot's capability to attribute mental states to the user and build a profile of her. We define *user's model* $\mathcal{M}_H = (\mathcal{B}_H, \mathcal{G}_H, \Pi_H)$ as a set of beliefs $\mathcal{B}_H$, goals $\mathcal{G}_H$ and plans $\Pi_H$. Each belief $\beta_i \in \Omega_R$ is represented and refers to the features defining the user's profile which are perceivable by the robot itself. This representation of beliefs determines *stereotypes* [65], useful for the robot to activate a task (for example, if the user likes modern art, the robot will typically propose a modern art collection). Given $\mathcal{P}_H$, the architecture allows us to select the most suitable user's model $\mathcal{M}_H$; in particular, each belief $\beta_{ms_{Hi}} \in \mathcal{M}_H$ is selected on the basis of a measure of *suitability* $S_\beta$. The suitability of a belief quantifies how much the belief is adapted to a specific type of user on the basis of the profile attributed to her. Let us give an example of building the user's model: a robot plays the role of museum assistant, and it has the goal to provide to the user a personalized museum exhibition to visit. During an initial interaction with a user, the robot retrieves perceptual information that allows it to infer that the user is young, with a high level of education, likes impressionism and hates ancient art. These features collected in the profile are integrated in the robot's beliefs base. The robot exploits the inferred profile in order to attribute specific behaviors and mental states to the user, leveraging the suitability value associated with each belief represented in its own beliefs base.

### 3.3.2. Active Goals Selection

The effective process leading the robot to adopt the final task starts after it defines $\mathcal{P}_H$. The complex module of *Reasoning Task Adoption* implements an algorithm that infers goals and plans of the user after she has (directly or indirectly) delegated a task. Because of their definition in $\Omega_R$ (see Section 3.1), plans can be considered hierarchical constructs of elementary or complex actions. Consider Figure 2, part (a): the same task $(\alpha_0, g_0)$ represents the node of different complex plans, which involve the achievement of several other actions/goals. If the robot decides to over-help the user by choosing a much more complex plan/goal than the delegated one (delegated task is still part of the new complex one), it has to take into account the model $\mathcal{M}_H$ in order to select those goals/actions that are suitable for her. Furthermore, let us consider part (b) of the same figure: the same goal $(g_1)$ is achieved by two different plans. If the robot decides to adopt a critical help strategy by choosing a different plan than the delegated one, which allows it to achieve the goal delegated by the user, it has to take into account the model $\mathcal{M}_H$ in order to select those plans that are suitable for the user herself.

We define *active goal* $g_A$ as a goal that the user, with a specific profile and set of mental states, did not delegate but that might represent a state of affairs she could be interested in or has already planned. In practice, these additional goals or interests of the user (active goals) are deduced on the basis of the profile that the robot builds and the consolidated knowledge on potential categories. Given the set of plans $\Pi \in \Omega_R$ and the user's profile $\mathcal{P}_H$, the *Active Goals Plans Selection* module allows us to enrich the model $\mathcal{M}_H$ with further goals (in addition to the delegated one) by sorting the plans $\pi_j \in \Pi(j = 1, .., s)$ on the basis of their corresponding value of suitability $S_{\pi_j}$. If two or more plans have the same suitability, they are also sorted on the basis of the highest number of goals they satisfy.

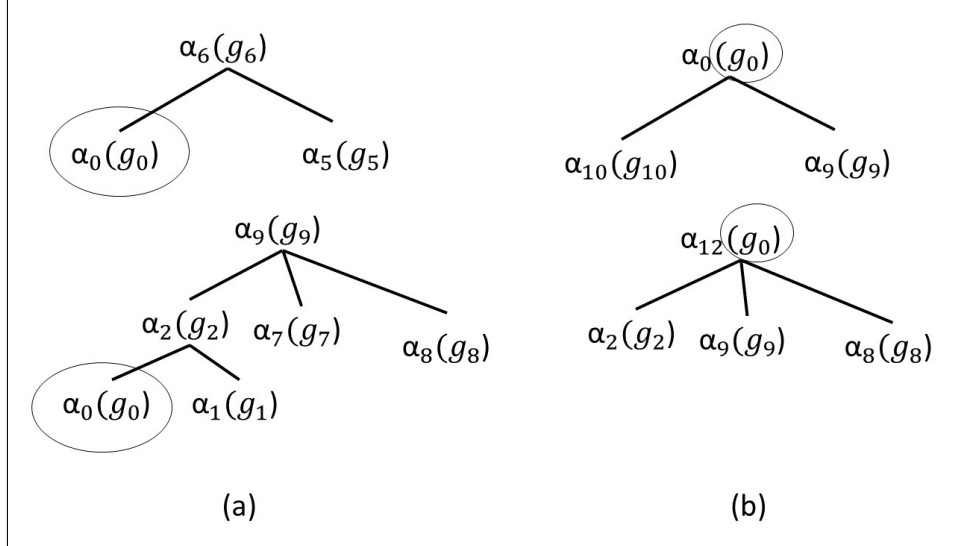

**Figure 2.** Example of plans in $\Omega_R$ exploited by the robot in order to perform respectively over help (**a**) and critical help (**b**).

### 3.4. The Task Selection Process

A crucial property that promotes intelligent help is the capability to apply the right level of help with which the robot intends to adopt the delegated task. Intelligent collaboration expresses the ability to consider the literal user's request as an element to integrate with other contextual elements (i.e., user needs, physical environment). Sometimes, the interpretation of these elements implies risks of errors and failures; sometimes, although correctly assessed, they can lead to an excess of collaboration that is not necessarily appreciated by the recipient. For doing that, we consider different robot *dispositions* that correspond to different levels of *proactiveness* with which the robot intends to achieve the delegated task. In particular, we define levels of a robot's proactiveness, *literal* and *over*. The first level allows the robot to try to achieve the goal delegated by the user (at most by changing the plan and performing critical help); the over level allows the robot to carry out a much more complex plan/goal than the delegated one of which the delegated task is still a part. Each robot's disposition corresponds to a specific heuristic for task selection, (collected in the module *Heuristics for Task Selection*) that implement different task selection algorithms (based on the two level of proactiveness defined above) through which the robot adjusts its own level of collaborative autonomy by restricting or expanding the delegated task on the basis of the mental states it has represented in its own Beliefs Base (see Section 3.1). The *sub-help* level defined in Section 2 occurs every time the robot fails to accomplish the task defined in the literal or critical help levels, but at least a sub-goal of $g_0$ is achieved.

In this work, we provide results of an experiment where the heuristic exploited by the robot just allows it to perform *literal* or *critical* help. (We do not consider the case of over-help). This means that the robot tries to adopt exactly what has been delegated by the user (performing literal help), but when it believes that there is a more suitable way for the user to achieve the same goal (based on the mental states attributed to the user and those attributed to other agents involved in the interaction), the robot changes the user's plan, performing critical help. In Section 4, we will describe the heuristic exploited by the agent in the pilot study. As mentioned, the use of this heuristic does not exclude the presence of other heuristics. For example, the robot can be equipped with other heuristics that allow it to perform over-help whenever it believes there is the context to go beyond what has been delegated by the user without changing her plan but including it within a hierarchically superior plan, which includes active goals not declared by the user. The common thing of every heuristic is to make a decision based on the mental states attributed to the user, always trying to satisfy her goals or needs as much as possible, also taking into consideration the presence of other agents involved in the interaction.

*3.5. The Task Adoption Explanation*

After the robot completes the task and provides the results to the user, it can provide an explanation about the why it adopted a specific strategy for achieving the delegated task in that way. The robot's reasons provided for justifying the task adoption are the mental states exploited during each phase of the perception, reasoning and action process implemented by the cognitive architecture. In particular, the explanation includes a mechanism to construct a behavior *log*, to which the robot attributes beliefs or goals to the user, on the basis of its capability to profile the user and to have a ToM both of the user and of other agents involved in the interaction. During the reasoning process implemented by the architecture, the robot records information (encoded as beliefs) that will be useful for building the explanation.The behavior log is the flowchart structure represented in Figure 3: initially the robot describes the task, then it describes the relevant beliefs it has attributed to the user based on its capability to profile her and to represent her mental states and that have been crucial for the robot's choice. In this phase, the robot also describes (if they exist) the active goals attributed to the user. After that, the robot introduces its disposition which corresponds to the levels of proactiveness defined in Section 3.4.

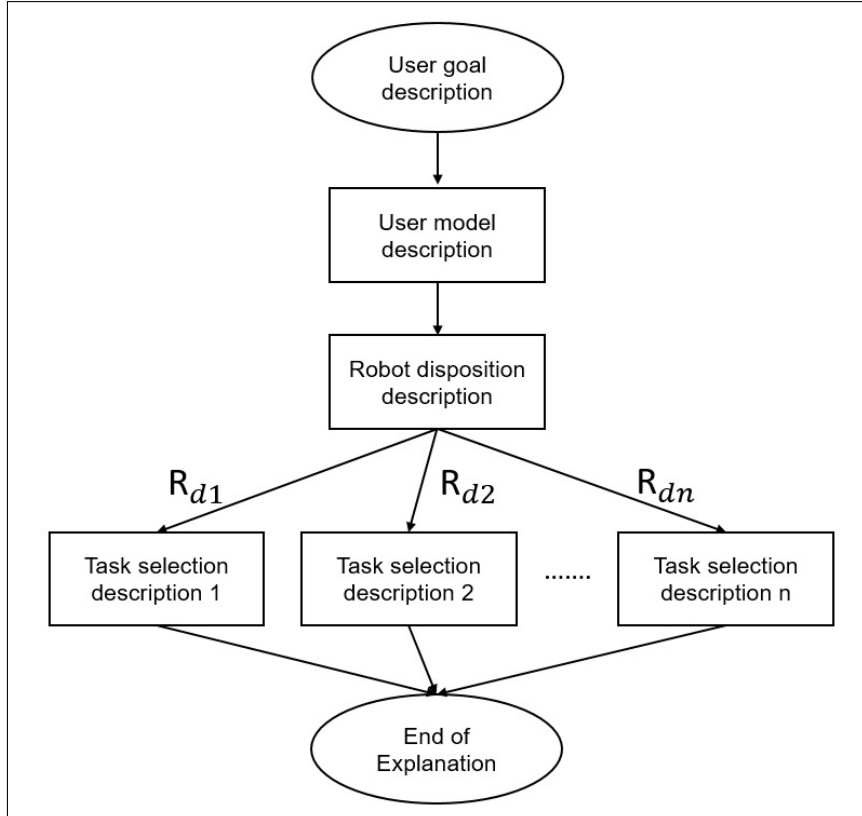

**Figure 3.** Flowchart exploited by the robot in order to provide an explanation about the reasons that led it to complete the task.

Based on the heuristic implemented, the robot exploits the beliefs that it has logged during the task selection to provide the explanation of the criteria with which it has chosen the final task, regardless of whether it coincides with the task delegated by the user. In the experiment section (Section 4), we provide an example of the explanation produced by the robot based on the heuristic exploited by the robot in the experiment.

**4. The Pilot Study**

In order to test the potentiality of our computational model, we ran a human–robot interaction pilot study. A total of 26 participants, aged between 25 and 75, were recruited for this pilot study. Each participant: (i) has interacted with the humanoid robot Nao; (ii)

carried out an entire interaction with the robot (*trial*); (iii) has the goal to visit (delegated task) the virtual museum, corresponding to a specific artistic period; and (iv) is aware of the fact that the tour will be chosen by the robot that manages the virtual museum, who will choose the most suitable tour for her. In this scenario, the robot is a museum assistant that handles a virtual museum, and it provides a museum tour (adopted task) to the user.

The virtual museum is organized in multiple thematic tours (Figure 4), each containing artworks (Figure 5) of the same artistic period (e.g., Impressionism, Surrealism, Baroque, Greek Art and so on). The museum is organized in such a way that it covers the entire body of the history of art. The categorization of the history of art periods follows the schema shown in Figure 6 and it is based on the work of one of the most important art historians of the 20th century, Giulio Carlo Argan [66].

The museum is designed to establish potential assumptions believed by the users: for example the artistic periods belonging to the same category are more homogeneous and therefore corresponding in the preferences of the users with respect to artistic periods of other categories. A user that indicates as the preferred artistic period "Impressionism" probably will be more inclined to "modern art" rather than "ancient art".

Three attributes describe a thematic tour: *Relevance*, *Accuracy*, and *Category*.

- The *Relevance* of an artistic period is defined on the basis of the originality of the artworks that compose it and the impact they had in the field of art history.
- The *Accuracy*, on the other hand, specifies the detail in the description of each artwork present in a thematic room.
- Each thematic tour (artistic period) belongs to a *Category*, which collects different artistic periods; for example, the "Impressionism" tour belongs to the same category as the "Surrealism" and "Cubism" tour, which are in the more general class named "modern art". This is replicated for any artistic period.

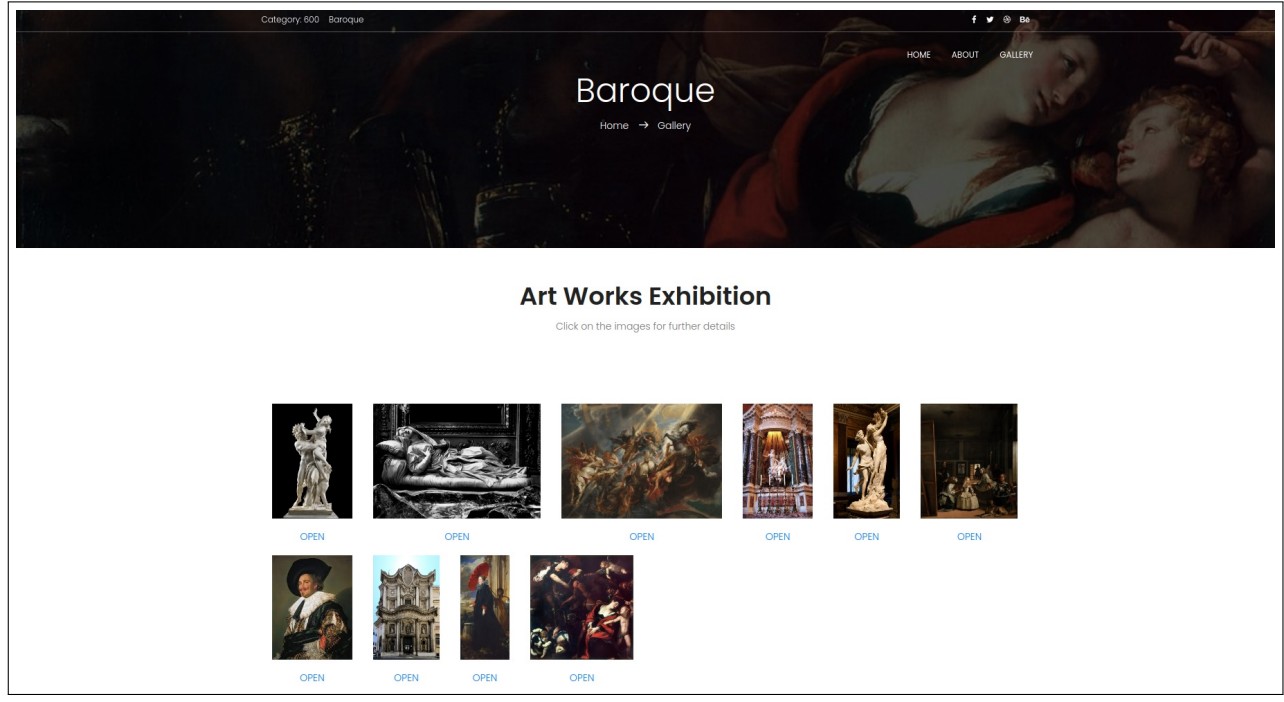

**Figure 4.** A thematic tour is a collection of artworks of the same artistic period (i.e., Baroque). By clicking on *OPEN*, it is possible to access a more detailed description of the artwork itself.

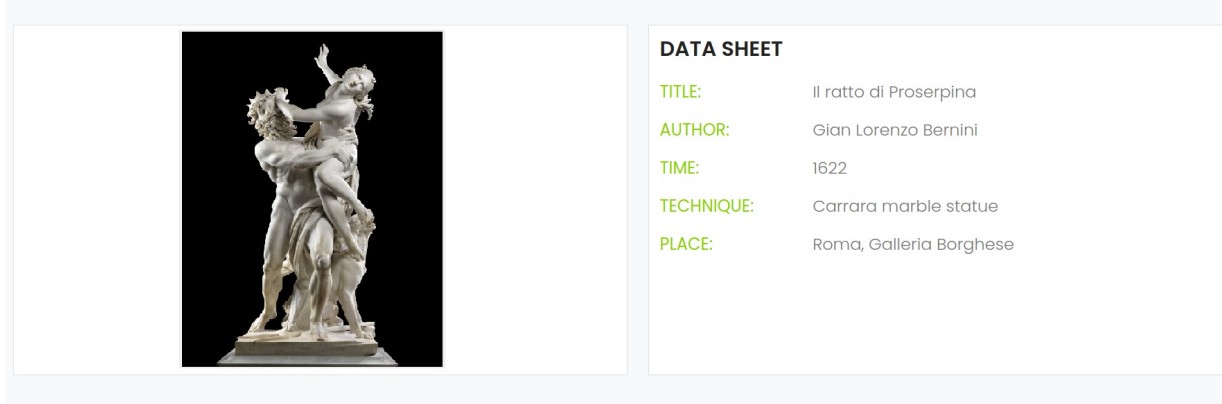

**Figure 5.** Example of artwork. On the left is a copy of the artwork, the right part contains the artwork's data sheet. If the level of accuracy is high, the user can read, in addition to basic information such as title, author, time, technique, place, a more detailed description. The description can be read by the robot (by clicking on *HERE* button), as well as by the user. If the level is low, the bottom description is excluded. (Image credits : Galleria Borghese/photo Luciano Romano).

| Ancient Art | • Origins & Civiltà Egea<br>• Greek Art<br>• Etruscan Art<br>• Roman Art |
|---|---|
| High Middle Ages | • Early Christian & Byzantine Art<br>• Romanic |
| Low Middle Ages | • 300<br>• Italian & International Gothic |
| Renaissance | • 400 Painting<br>• 400 Architecture<br>• 400 Sculpture<br>• 500 of Michelangelo, Leonardo & Raffaello<br>• 500 of Tiziano & Giorgione<br>• Manierism |
| 600 | • Caravaggio & Caravaggieschi<br>• Baroque |
| 700 | • 700 Painting<br>• 700 Architecture<br>• 700 Sculpture |
| Modern Art | • Romanticism<br>• Neoclassicism<br>• Impressionism<br>• Simbolism<br>• Art Nouveau<br>• Espressionism<br>• Cubism<br>• Surrealism<br>• Dadaism |
| Contemporary Art | • Contemporary Art |

**Figure 6.** History of Art categorization exploited for designing the virtual museum.

In this experiment, we consider three levels of relevance (high, medium, low) and two levels of accuracy (high, low). The user can explore the museum room by choosing the artwork she wishes, and she can leave the museum at any time.

Each trial is organized in four phases:

1. *Starting Interaction*: The robot introduces itself to the user, describing its role and the virtual museum it manages.
2. *User artistic profiling*: The robot asks the user a series of questions which aim to investigate her artistic interests in terms of her favorite artistic periods and artistic periods of no interest. In this phase, the interaction is supported by a GUI through which the user can express her artistic preferences, and the robot can collect useful data to profile the user. In addition to defining the artistic periods of interest and non-interest of the user, the robot asks the user with what degree of accuracy she intends to visit the section.
3. *Tour visit*: Once the user profile has been established, the robot exploits the heuristic defined in the Section 4.1 to select the tour on behalf of the user. Once the selection has been made, the robot activates the corresponding tour in the virtual museum and leaves the control to the user who can visit the room, selecting the artworks inside.
4. *End museum tour*: The user can leave the recommended tour and therefore the museum. Once this happens, the robot returns to interact with the user, asking her questions. These questions, which belong to a short survey, are used to investigate how satisfied the user was after the visit.
5. *Explanation phase*: The robot provides to the user an explanation of its behavior in order to justify the reasons that lead it to recommend that specific museum tour. After the explanation, the robot asks the user if she wants to answer the short survey administered in the previous phase again.

In particular, the survey's questions that the user had to answer are the following:

- Q1: How satisfied were you with the duration of the visit?
- Q2: How satisfied were you with the quality of the artworks?
- Q3: How satisfied were you with the number of the artworks?
- Q4: How surprised were you with the artistic period recommended by the robot compared to the artistic period you initially chose?
- Q5: How satisfied are you with the robot's recommendation given the artistic period you initially chose?

Each participant has been administered the same questions, rated using a 5-point Likert scale [1 = worst case and 5 = best case]. For example, considering the question Q5, the value 1 means that the user was completely dissatisfied with the robot's recommendation given the artistic period initially chosen, value 5 means that the user wasn completely satisfied by the robot's decision. The same goes for the other questions. The participants communicate any answer to the robot by voice. We have not given the users any prior instructions. Each instruction was given by the robot during the entire interaction with the user.

### 4.1. The Heuristic for the Tour Selection

The Algorithm A1 in Appendix A describes the heuristic exploited by the agent in order to select the most appropriate section to visit. The algorithm takes as input the user-preferred artistic period $(p_F)$, the periods of non-interest $(P_D)$ and the level of accuracy chosen by the user to visit the exhibition $Acc_u$. After obtaining the values of relevance $(r_{p_F})$, accuracy $(a_{p_F})$ and the category $(c_{p_F})$ of the tour corresponding to the user preferred artistic period, the algorithm checks multiple conditions. The algorithm allows us to discard artistic periods that are not of interest to the user as well as the favorite period. The resulting set $(P_M)$ represents the set of artistic periods different from the preferred one, which continue to be of interest to the user. The first condition (condition $C_1$) requires us to verify if the same artistic period required by the user has maximum relevance from the museum curator's point of view $(r_{Max})$ and if the accuracy of its description corresponds with that chosen

by the user ($a_{p_F} = Acc_u$). If these two conditions are true, then the robot will recommend the visit of the corresponding tour ($R_{toVisit}$). If just the accuracy condition is not satisfied, however, the algorithm chooses the period required by the user and presents it with a level of accuracy different from that indicated. The accuracy will be the one believed by the museum curator (condition $C_2$). If condition $C_2$ is not verified either, then the algorithm investigates the tours corresponding to the artistic periods that the user has not discarded ($P_M$). If there is a tour with a high level of relevance which belongs to the same category as the user-preferred artistic period and which requires a level of accuracy equal to that chosen by the user, then the robot will recommend the visit of the corresponding tour (condition $C_3$). If not even this condition is verifiable, then the algorithm will try to select a tour with a high level of relevance which belongs to the same category as the user-preferred artistic period, regardless of the level of accuracy it requires; the accuracy will be the one believed by the museum curator (condition $C_4$). Condition $C_5$ instead occurs when, having not found any tour to recommend in the same class in which the required artistic period was contained, there is a tour that corresponds to an artistic period belonging to the next or previous category to the user-preferred artistic period and has a level of relevance immediately following that of the user-preferred artistic period. Finally, if even $C_5$ is not acceptable, then the algorithm selects a random tour among those corresponding to the artistic periods not discarded by the user (condition $C_6$).

## 5. Results

In the pilot, three research questions (**RQ**) have been investigated:

- **RQ1:** How risky/acceptable is the critical help compared to the literal help? Does the heuristic proposed help make this help much more acceptable?
- **RQ2:** Given the risks that the critical help in any case determines, in what situations and how much critical help can be useful?
- **RQ3:** How much does the capability of the robot to explain why it perform critical-help affect the acceptability of the critical help itself?

This study has been organized as a between-participants experiment. As mentioned before, we recruited 26 real participants in total, which would allow us to detect an effect size of $d \geq 1$ with at least 0.8 power at an alpha level of 0.05 (calculated with the Jamovi software [67]), which is a reasonable value for an experiment [68].

Tables 1 and 2 contain, respectively, the survey question answers provided by the users who have received literal help from the agent (the preferred artistic period selected by the user coincides with the tour recommended by the robot) and those answers given by the users who have received critical help (the preferred artistic period chosen by the user does not match with the tour recommended by the robot). The agent provided literal help to 11 users, while 15 users received critical help. Among all the answers given by users, we focused on those related to the questions Q4 and Q5; these questions allow us to investigate what has been the impact of the robot's ability to propose to the user a tour in fact different from what she expected. The answers to questions Q1, Q2 and Q3 have not been analyzed. Their function in the survey will be explained in Section 5.2.

Table 3 shows the results obtained by running a parametric analysis of the answers to question Q5, provided by all the users involved in the pilot study. We can observe that users who have received literal help show a level of satisfaction ($M = 4.36$) on average higher than that of the users to whom the robot performed critical help ($M = 3$). Thus, critical help implies the risk of leaving the user at least partially dissatisfied because the robot expressly violated her requests. However, although the difference between the averages in the two cases is significant ($D = 1.36$), the mean value of the satisfaction referred to critical help highlights how this type of help does not raise a low level of satisfaction. Indeed, the value three in the scale exploited for the survey, corresponds to a medium level of satisfaction. This result is relevant if we consider that no justification has been provided by the robot for its behavior in contracting with the user requests. Due to the non-normal data

distribution, we ran a confirmatory Mann–Whitney test on the same data exploited in the *t*-test. The *p*-value associated to the non-parametric analysis is $p = 0.007$.

**Table 1.** The table summarizes the answers provided by the users to the questions in the survey after the robot provided literal help to the user. The robot recommended a tour corresponding to the artistic period the user indicated as preferred in the history of art.

| User | Preferred Artistic Period | User Accuracy | Recommended Tour | Tour Accuracy | Q1 | Q2 | Q3 | Q4 (Surprise) | Q5 (Satisfaction) |
|------|---------------------------|---------------|------------------|---------------|----|----|----|---------------|-------------------|
| 2 | 500′ Italian Painting | High | 500′ Italian Painting | High | 4 | 5 | 4 | 1 | 5 |
| 5 | 500′ Italian Painting | High | 500′ Italian Painting | High | 5 | 5 | 5 | 2 | 5 |
| 6 | Greek Art | Medium | Greek Art | Medium | 5 | 5 | 5 | 1 | 5 |
| 7 | Gothic | Medium | Gothic | Medium | 5 | 5 | 5 | 1 | 4 |
| 9 | 500′ Italian Painting | Medium | 500′ Italian Painting | High | 3 | 4 | 4 | 2 | 3 |
| 11 | Caravaggio | Low | Caravaggio | High | 5 | 4 | 4 | 1 | 4 |
| 13 | Gothic | Low | Gothic | Low | 1 | 1 | 1 | 1 | 2 |
| 14 | Contemporary Art | Medium | Contemporary Art | Medium | 5 | 3 | 3 | 1 | 5 |
| 16 | 700′ Painting | High | 700′ Painting | Medium | 4 | 4 | 5 | 2 | 5 |
| 18 | 500′ Italian Painting | High | 500′ Italian Painting | High | 5 | 3 | 4 | 1 | 5 |
| 19 | Contemporary Art | High | Contemporary Art | Low | 5 | 4 | 4 | 1 | 5 |

**Table 2.** The table summarizes the answers provided by the users to the survey questions after the robot provided critical help to the user. The robot recommended a tour different to the preferred user's artistic period.

| User | Preferred Artistic Period | User Accuracy | Recommended Tour | Tour Accuracy | Q1 | Q2 | Q3 | Q4 (Surprise) | Q5 (Satisfaction) |
|------|---------------------------|---------------|------------------|---------------|----|----|----|---------------|-------------------|
| 1 | Baroque | Medium | Caravaggio | High | 4 | 5 | 4 | 5 | 5 |
| 3 | Baroque | High | Caravaggio | High | 4 | 5 | 5 | 4 | 4 |
| 4 | Impressionism | Medium | Romanticism | Medium | 5 | 4 | 4 | 5 | 3 |
| 4 | Cubism | High | Espressionism | Medium | 3 | 2 | 3 | 5 | 1 |
| 5 | 700′ Sculpture | High | 700′ Painting | High | 5 | 5 | 4 | 3 | 4 |
| 8 | Cubism | High | Neoclassicism | High | 5 | 4 | 4 | 4 | 3 |
| 10 | Impressionism | High | Espressionism | Low | 4 | 1 | 3 | 5 | 1 |
| 12 | Impressionism | High | Surrealism | Medium | 3 | 3 | 4 | 3 | 4 |
| 15 | Art Nouveau | Medium | Romanticism | Medium | 5 | 5 | 5 | 4 | 3 |
| 17 | Art Nouveau | Medium | Neoclassicism | High | 5 | 5 | 4 | 3 | 4 |
| 20 | Futurism | Medium | Romanticism | Low | 4 | 2 | 4 | 5 | 1 |
| 21 | Cubism | Medium | Surrealism | Medium | 5 | 4 | 3 | 3 | 3 |
| 22 | Baroque | High | 400′ Painting | High | 2 | 5 | 3 | 4 | 1 |
| 23 | Romanticism | Medium | Simbolism | High | 1 | 5 | 4 | 3 | 4 |
| 24 | Cubism | Medium | Surrealism | Medium | 5 | 3 | 5 | 5 | 4 |

**Table 3.** Independent Samples T-Test run in order to evaluate the risk/acceptability of the critical help compared to the critical help (**RQ1** ). The *p*-value associated to the T-Test is $p = 0.010$; the Cohen's d is $d = 1.11$ that represents a large effect size.

| Group | Literal Help | Critical Help |
|-------|--------------|---------------|
| Mean | 4.36 | 3.00 |
| SD | 1.03 | 1.36 |
| N | 11 | 15 |

Figure 7 focuses on the group of participants for whom the robot has performed critical help (Table 2). The pie chart shows that 27% of participants evaluated the recommended tour completely unsatisfactory, 27% of the them evaluated it with a medium level of satisfaction, while 46% of them evaluated the tour satisfactory; of this 46%, 40% rated it satisfactory and 6% very satisfactory. If we consider any satisfaction value greater than or equal to three as a positive evaluation of the robot's decision, we can conclude that 73% of participants who received critical help by the robot evaluate this choice positively.

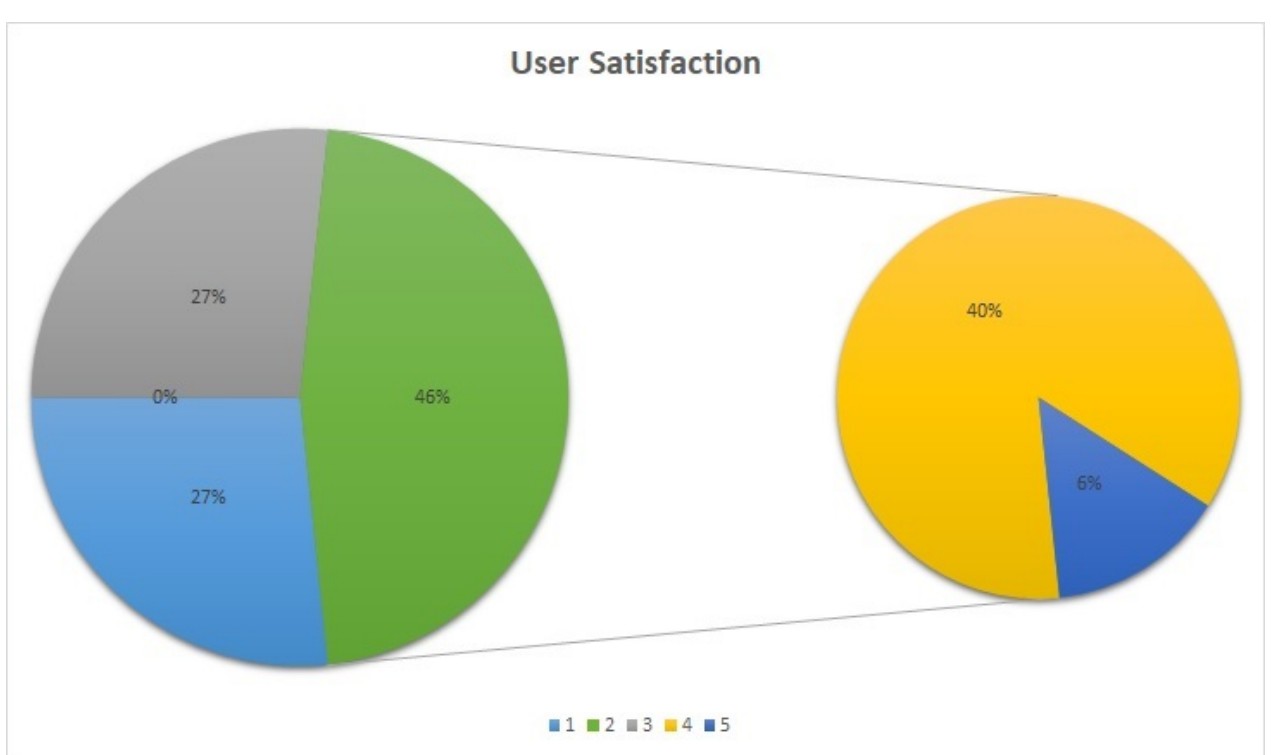

**Figure 7.** The pie chart reports the distribution of user satisfaction investigated through question Q5 in the case of the robot providing critical help to the participants.

This result is relevant if we consider that the participants were not instructed about the possibility that their request could be changed by modifying the artistic period chosen. Any prior information is communicated to the participant, and it is difficult for her to directly deduce it. The surprise effect encoded by the answer to question Q4, confirms this unexpected choice of the robot in the face of an explicitly different request.

In conclusion, we present the results related to the research question **RQ3**. As mentioned above, Tables 1 and 2 summarize the results obtained during the experiment, referring, respectively, to the users who received critical help and those who received literal help. Each user listened to the robot's reasons in the explanation phase. After the robot explanation, only some users considered it appropriate to respond again to the survey administered by the robot at the end of the museum tour visit. Table 4 shows the users who replied to the robot survey and the answers given before and after the explanation.

The rows of the table highlighted in gray refer to users who have received critical help from the robot, while the others refer to users who have received literal help. We can see how six users in total decided to answer the questions proposed by the robot again; three users received literal help, three users received critical help. The users who received literal help did not change the evaluation of their level of satisfaction/surprise, while two-thirds of the users who received critical help changed the evaluation of their satisfaction. We can observe how in each of the two cases this evaluation corresponds to an increase in satisfaction.

Figure 8 shows the average of the satisfaction values evaluated before and after the explanation, respectively, in the case of literal help (bar plot 1 on the left) and in the case of critical help (bar plot 2 on the right).

**Table 4.** This table reports the answers to the questions in the survey posed by the robot before and after the robot has provided the explanation about the task adoption. The rows highlighted in gray refer to the cases in which the robot provided critical help to the user. Other rows indicate the case in which the robot provided literal help to the user .

| User | Preferred Artistic Period | User Accuracy | Recommended Tour | Tour Accuracy | Q4 (Surprise) | Q5 (Satisfaction) | Explanation |
|------|---------------------------|---------------|------------------|---------------|---------------|-------------------|-------------|
| 2 | 500' Italian Painting | High | 500' Italian Painting | High | 1 | 5 | Before |
| 2 | 500' Italian Painting | High | 500' Italian Painting | High | 2 | 5 | After |
| 3 | Baroque | High | Caravaggio | High | 4 | 4 | Before |
| 3 | Baroque | High | Caravaggio | High | 4 | 4 | After |
| 9 | 500' Italian Painting | Medium | 500' Italian Painting | High | 2 | 3 | Before |
| 9 | 500' Italian Painting | Medium | 500' Italian Painting | High | 2 | 3 | After |
| 10 | Impressionism | High | Espressionism | Low | 5 | 1 | Before |
| 10 | Impressionism | High | Espressionism | Low | 4 | 4 | After |
| 16 | 700' Painting | High | 700' Painting | Medium | 2 | 5 | Before |
| 16 | 700' Painting | High | 700' Painting | Medium | 2 | 5 | After |
| 21 | Cubism | Medium | Surrealism | Medium | 3 | 3 | Before |
| 21 | Cubism | Medium | Surrealism | Medium | 5 | 4 | After |

All users who received critical help either maintained the same level of satisfaction (user 3) or increased it, in one case even considerably (user 10). Furthermore, although there were cases (user 21) in which the evaluation of the surprise increased, this increase did not have negative repercussions on satisfaction; on the contrary it resulted in an increase. Conversely, when the surprise decreased (user 10), the satisfaction increased considerably. We can conclude that if all users completed both the surprise and satisfaction evaluation, the satisfaction ratings would have been higher.

These results, although preliminary and based on a small sample, seem promising in that they show the impact that the explanation of the task adoption strategy had on the acceptance of critical help, in which the user's requests were not completely fulfilled.

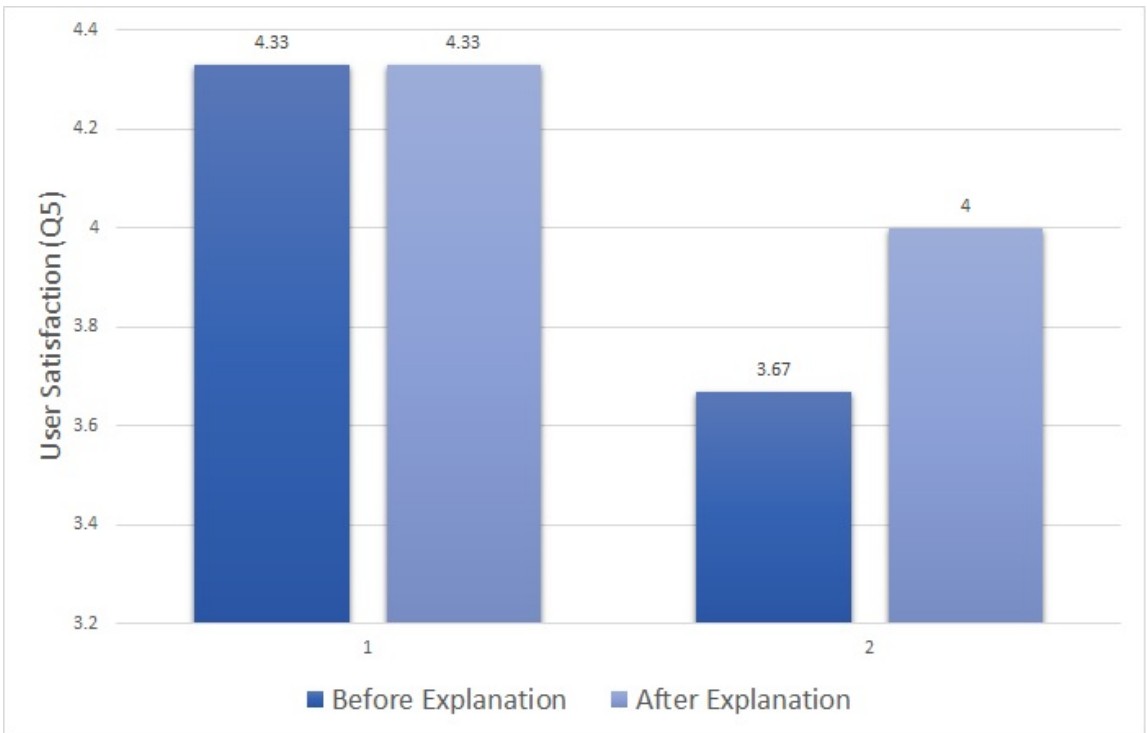

**Figure 8.** The bar plot reports the mean values related to the answer to Q5 before and after the user received an explanation from the robot in the case of literal help (**left** side) or critical help (**right** side).

### 5.1. Robot Explanation Example

Here we propose an example of the explanation proposed by the robot to user 21 (see Table 4). The explanation is based on the algorithm represented in Figure 8, and it is based on the heuristic described in Section 4.1.

The Figure 9 shows the explanation flow provided by the robot to the user after the visit to the museum and after the user answers the survey. The robot starts by introducing the task delegated by the user, which in this case is *visiting the Cubism tour*. After that, the robot explains to the user some features of her model that the robot itself created in its own belief base based on the user's profile, which the robot was able to create thanks to an initial interaction that investigated her artistic interests, both the favorite (Cubism) ones and those of no interest (Greek Art, Roman Art, 600 s). The robot continues the explanation by describing its level of proactiveness. In this case, the robot meets the user's goals by providing literal or critical help; the results derive from an internal mediation process that takes into account both the user's goals and the goals of the museum curators who have designed the exhibitions. Furthermore, the robot exploits the reasoning process defined by the heuristics introduced in Section 4.1 in order to explain the reasons that led it to suggest the visit of the Expressionism tour rather than the Cubism tour. In reference to the heuristic, the robot exploits condition $C_3$: the Cubism tour has low relevance for the museum curators, so the robot looks at a tour with a high level of relevance, which belongs to the same category as the user-preferred artistic period (in this case modern art) and which requires a level of accuracy equal to that chosen by the user (in this case medium accuracy). Finally the robot recommends the visit of the corresponding tour. In this case, the result of the selection is the Expressionism tour.

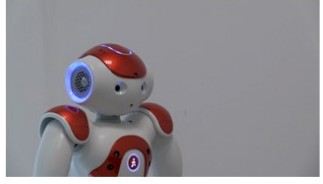

I explain to you the reasons that led me to suggest you the visit of the *Expressionism* museum tour.

↓

Based on the initial interaction with you, I found that you are particularly passionate about *Cubism*. On this basis, I believe you are a person who prefer *modern art*. Conversely, I believe that you are disinterested in ancient art and the art of the 600s. Finally, I have found that the level of accuracy with which you intend to visit the tour is *medium.*

↓

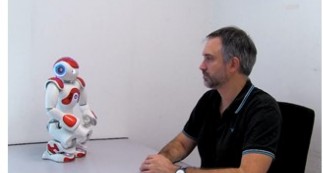

I am a robot who tries to achieve the goals of every visitor, but I also take into account the goals of those who designed the museum tours. So, if the tour that I have suggested is different from your expectations, it is precisely because of that. The tour you have chosen could be judged by its curators with a not very high relevance.

↓

The *Cubism* tour has low relevance for the museum's curators, which means that there no relevant works inside the tour. I chosen for you a tour that is judged by the curators to be very relevant and that requires the same level of accuracy as you choose. Since I believe you love *modern art*, I have chosen for you the *Expressionism* tour, which, unlike Cubism, is very relevant from an artistic point of view. Here you will find very important works.

↓

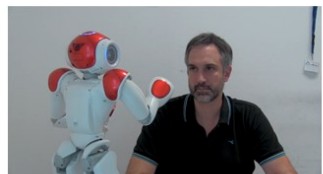

I hope my explanations have been useful to you in judging my help. Thank you

**Figure 9.** Example of the explanation provided to user 21.

*5.2. Experiment Limitations*

The pilot study presents a series of limitations. First of all, the target sample size is small. Despite that, the results we analyzed in Section 5 allow us to make different considerations about the robot's capability to provide adaptive and intelligent help every time it has to perform a task for a user. Furthermore, the participants have been recruited without considering specific criteria, such as their artistic habits or their previous experience with robots, their comfort, and willingness to interact and accept the introduction of robots in society. Especially with respect to the robot acceptability, we tried to reduce this bias by tailoring the survey proposed by the robot in such a way that the user was forced to answer preliminary questions (Q1, Q2, Q3) designed to focus the user on the content of the tour and not only on the interaction with the robot. Finally, we built the survey with ad hoc questions. This limitation can represent an obstacle in comparing our work with other similar works. Although this preliminary experiment has a number of limitations, the obtained results shown some significance. Because of that, we will try minimize the biases discussed above by following much more standard methodological approaches [69–71].

**6. Conclusions and Future Works**

Over time, collaboration between human beings, one of the fundamental characteristics of social development, has evolved and has been able to rely on more and more advanced technological supports and tools. The collaboration is deep so it takes into account not only what is explicitly declared among participants but also the implicit contexts of expectations and of a whole series of other factors. The creation of artificial autonomous systems, such as robots and more generally intelligent agents, has moved collaboration forward compared to the old technological supports of collaboration. Having their own real spaces of autonomy allows these systems to fulfill their collaborative tasks in a much more advanced way and in the direction of intelligent collaboration typical of humans: the only one that has that added value that can provide intelligence. In this work we have presented a cognitive architecture for social robots, and we showed its implementation and a pilot study that highlights its potential for effective collaboration. The architecture developed refers to consolidated theoretical principles: theory of adoption and delegation, ToM, theory of social adjustable autonomy. The architecture gives a robot the capability to

build a profile of the user and to exploit its profiling skills to attribute beliefs, goals, plans, and intentions of the user and makes it capable of modulating its autonomy for completing a delegated task. One of the main problems in intelligent collaboration between humans is the possibility of misunderstandings that can lead to conflicts between collaborators. We call these conflicts collaborative conflicts, as they are based on the desire to collaborate beyond what is required but in doing so introduce errors and discrepancies. Explaining the action/plan helps minimize these conflicts among collaborators. With autonomous intelligent systems, this need to access explainability is becoming a major study requirement because the new learning algorithms (deep learning) introduce several problems from this point of view. In intelligent collaboration, this element of explanation becomes even more pressing, all the more if created by an autonomous robotic system.

The cognitive architecture was tested in the domain of cultural heritage. The exploratory HRI study showed promising results. Because of that, the main future work consists of a follow-up on this pilot study to systematize the preliminary results obtained and to give consistency to the research questions we investigate. The experiment will try to overcome all the limitations introduced in the pilot study, starting from the sample of users. Furthermore, we intend to extend the computational model by integrating other levels of help, as provided by the delegation and adoption theory and to test their impact through other HRI experiments in real scenarios.

## 7. Related Work

Cultural heritages sites such as museums represent a complex scenario, suitable for robots that are able to assist and interact with people in a natural manner. Different pioneering work [72–76] has been proposed, with the goal of designing robots able to be deployed in a museum and to integrate different human–robot interaction capabilities. Despite their pioneering and remarkable work in autonomous navigation and human–robot interaction, these approach do not realize a real *Personalization* [9] of the user experience. Recent works [4,5,77] try to implement a much more effective and personalized user experience by designing robots that are able to exploit their perceptive and decision-making skills in order to establish a much more deep interactions with visitors However, these approaches do not take into account the mental states of the interacting user; therefore, they do not achieve a real personalization [78] based on complex models.

**Author Contributions:** Conceptualization, methodology, validation, formal analysis, investigation, resources, data curation, writing—review and editing: F.C. and R.F.; software, writing—original draft preparation, visualization: F.C. All authors have read and agreed to the published version of the manuscript.

**Funding:** This research received no external funding.

**Acknowledgments:** We would like to thank *Galleria Borghese*, in Rome, for having granted the publication of a photograph of the artwork *il ratto di Proserpina* (photo Luciano Romano). Furthermore, we would like to thank Cristiano Castelfranchi for the precious suggestions and discussions on this work.

**Conflicts of Interest:** The authors declare no conflict of interest.

## Appendix A

The appendix reports the heuristic exploited by the agent in order to select the most appropriate section to visit (Algorithm A1).

---

**Algorithm A1** Artistic Period Selection Algorithm

---

**Input:** $p_F, P_D, Acc_u$

1: **procedure** HEURISTIC FOR SELECTION
2: $\quad r_{p_F} \leftarrow getRelevance(p_F)$
3: $\quad a_{p_F} \leftarrow getAccuracy(p_F)$
4: $\quad c_{p_F} \leftarrow getCategory(p_F)$
5: $\quad P_M \leftarrow remove(P_D, p_F)$ $\qquad\qquad\qquad\qquad\qquad\qquad\quad \triangleright P_M$ contains all section
6: $\quad$ **if** $(r_{p_F} = r_{Max}$ & $a_{p_F} = Acc_u)$ **then** $\qquad\qquad\qquad\qquad\quad \triangleright C_1$
7: $\qquad R_{toVisit} \leftarrow p_F$
8: $\quad$ **else**
9: $\qquad$ **if** $(r_{p_F} = r_{Max})$ **then** $\qquad\qquad\qquad\qquad\qquad\qquad\quad \triangleright C_2$
10: $\qquad\quad R_{toVisit} \leftarrow p_F$
11: $\qquad$ **else**
12: $\qquad\quad$ **for** $p_M \in P_M$ **do**
13: $\qquad\qquad r_{p_M} \leftarrow getRelevance(p_M)$
14: $\qquad\qquad a_{p_M} \leftarrow getAccuracy(p_M)$
15: $\qquad\qquad c_{p_M} \leftarrow getCategory(p_M)$
16: $\qquad\qquad$ **if** $(r_{p_M} = r_{Max}$ & $c_{p_M} = c_{p_F}$ & $a_{p_M} = Acc_u)$ **then** $\quad \triangleright C_3$
17: $\qquad\qquad\quad R_{toVisit} \leftarrow p_M$
18: $\qquad\qquad\quad$ **return** $R_{toVisit}$
19: $\qquad\qquad$ **else**
20: $\qquad\qquad\quad$ **if** $(r_{p_M} = r_{Max}$ & $c_{p_M} = c_{p_F})$ **then** $\qquad\qquad \triangleright C_4$
21: $\qquad\qquad\qquad R_{toVisit} \leftarrow p_M$
22: $\qquad\qquad\qquad$ **return** $R_{toVisit}$
23: $\qquad\qquad\quad$ **else**
24: $\qquad\qquad\qquad c_{next} \leftarrow getNextCategory(p_M, p_f)$
25: $\qquad\qquad\qquad c_{prev} \leftarrow getPreviousCategory(p_M, p_f)$
26: $\qquad\qquad\qquad r_{new} \leftarrow getNewRelevance(p_M, p_f)$
27: $\qquad\qquad\qquad$ **if** $(r_{p_M} = r_{new}$ & $(c_{p_M} = c_{next}|c_{p_M} = c_{prev}))$ **then** $\quad \triangleright C_5$
28: $\qquad\qquad\qquad\quad R_{toVisit} \leftarrow p_M$
29: $\qquad\qquad\qquad\quad$ **return** $R_{toVisit}$
30: $\qquad\qquad\qquad$ **else**
31: $\qquad\qquad\qquad\quad R_{toVisit} \leftarrow getRandom(P_M)$ $\qquad\qquad \triangleright C_6$
32: $\qquad\qquad\qquad$ **end if**
33: $\qquad\qquad\quad$ **end if**
34: $\qquad\qquad$ **end if**
35: $\qquad\quad$ **end for**
36: $\qquad$ **end if**
37: $\quad$ **end if**
38: **end procedure**

---

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
