# Peer review of "Collaborative Autonomy: Human–Robot Interaction to the Test of Intelligent Help"

_electronics, doi:10.3390/electronics11193065_

Round 1
Reviewer 1 Report
This paper presents a cognitive architecture for human robot interaction and evaluate the implemented method through the museum tour experiment.
Some parts of this paper are quite similar to the previous paper [13]. For example,
- L.597-615 Page 16-17 in the current paper (Figure 7 shows how, among ...) and the last paragraph Page 12 in the previous paper (Figure 5 shows how, among ...)
- Section 5.2 Page 19 in the current paper and Section Experiment Limitations Page 13 in the previous paper
- Section 6 Page 20 in the current paper and Section 6 and 7 Page 13-14 in the previous paper
- Section 7 Page 21 in the current paper and Section 1.1 Page 2 in the previous paper
I believe the authors were not willing to, but this might give a false impression. Please revise these parts of the manuscript for publishing as a paper.
How did the participants answer the questions? By writing number 1-5 on a sheet?
I guess it might be hard for the participants to answer their satisfaction. How did you instruct the participants?
Which independent t-test did you use for evaluation? Is there any assumption of the result data distribution?
What is the main functionality of the robot NAO? Did the agent need to be a humanoid robot?
Author Response
This paper presents a cognitive architecture for human robot interaction and evaluate the implemented method through the museum tour experiment.
Some parts of this paper are quite similar to the previous paper [13]. For example,
- L.597-615 Page 16-17 in the current paper (Figure 7 shows how, among ...) and the last paragraph Page 12 in the previous paper (Figure 5 shows how, among ...)
- Section 5.2 Page 19 in the current paper and Section Experiment Limitations Page 13 in the previous paper
- Section 6 Page 20 in the current paper and Section 6 and 7 Page 13-14 in the previous paper
- Section 7 Page 21 in the current paper and Section 1.1 Page 2 in the previous paper
I believe the authors were not willing to, but this might give a false impression. Please revise these parts of the manuscript for publishing as a paper.
Answer: First of all, we want to thank the reviewer for the interesting and stimulating suggestions and criticisms.
As stated and as noted by this reviewer, the present paper is a revised and enriched extension of a previous one ([13]). We have therefore transparently taken parts of that paper only to make the meaning of the extensions clearer. In this review, at the suggestion of the reviewer, we have reduced the overlap.
How did the participants answer the questions? By writing number 1-5 on a sheet?
Answer: The interaction between user and robot occurs mostly through the voice. We have thoroughly described in the paper how the interaction develops (see section 4).
I guess it might be hard for the participants to answer their satisfaction. How did you instruct the participants?
Answer: Participants are first told only that they will be guided by a robot in the visit of a virtual museum. All other information and requests are then carried out by the robot. Including the request to provide information on the satisfaction that the user felt both with respect to the visit (quality of the works of art), and with respect to the expectations on the way of collaborating that the robot has carried out.
Which independent t-test did you use for evaluation? Is there any assumption of the result data distribution?
Answer: We ran an independent samples t-test. Due to the non-normal data distribution, we ran a confirmatory Mann-Whitney test, on the same data exploited in the t-test.
What is the main functionality of the robot NAO? Did the agent need to be a humanoid robot?
Answer: There is an extensive literature that indicates that physical robots elicit more favorable social responses than virtual agents. (Li, Jamy. "The benefit of being physically present: A survey of experimental works comparing copresent robots, telepresent robots and virtual agents." International Journal of Human-Computer Studies 77 (2015): 23-37.). So in fact the use of an anthropomorphic robot (NAO or other kind) allows to make the interaction more natural and more stimulating for the user than compared to a virtual agent. We are aware that the robot NAO has technological limitations, but with this experiment we focus on the robot’s decision making process, that allow us to mitigate the tech limitations of the robot itself. In the future our idea is to move to much more complex robots that are able also to navigate autonomously in a unstructured space. But at this stage, the robot humanoid NAO is sufficient to carry out the task we want to pursue.
Reviewer 2 Report
The paper described architecture for Human-Robot Interaction. The author's background study is sufficient. The contribution to the field is clearly outlined in the paper. But I doubt whether the application scenario is really significant. Also, the participant's rating is subjective, which means the results may vary when changed participants. The scientific soundness need improvement. The appendix provided the algorithm, but the readability is relatively low. If the author can show their code, it would be better evaluated.
Author Response
The paper described architecture for Human-Robot Interaction. The author's background study is sufficient. The contribution to the field is clearly outlined in the paper.
But I doubt whether the application scenario is really significant.
Answer: First of all, we want to thank the reviewer for the interesting and stimulating suggestions and criticisms.
With respect to the significance of the application scenario described in the paper, our idea is that it is, both for the number of users used and for the experimental system realized, sufficient to investigate how an autonomous intelligent collaboration can be accepted/evaluated by human users. Obviously, characterized in the specific domain as that of support for visiting a museum.
Also, the participant's rating is subjective, which means the results may vary when changed participants. The scientific soundness need improvement.
Answer: This study has been organized as a between-participants experiment. We recruited 26 real participants in total, which would allow us to detect a large effect size (d>1) with an acceptable level of power (0.80) at an alpha level of 0.05. As we detected an even larger effect size (d=1.13), we can conclude that our test is reliable and sample size adequate to interpret our results. The independent samples t-test we ran has a p-value=0.001. Due to the non-normal data distribution we ran also a confirmatory Mann-Whitney test, on the same data exploited in the t-test. The p-value associated to the non-parametric analysis is p = 0.007. We are aware that the sample size is relatively low, but we exploited this exploratory experiment in order to make some assumptions and define some research questions. We think that the exploratory study results gaves us the support to follow up with a much more exstensive and sistematized experiment.
The appendix provided the algorithm, but the readability is relatively low. If the author can show their code, it would be better evaluated.
Answer: The pseudo-code shown in the appendix is certainly clearer than the experiment's source code.
We eliminate the first pseudo-code because we think that the description of the active goal selection algorithm is sufficient to get an overview of how the algorithm works. Furthermore, we enriched the description of the second algorithm (heuristic for task selection) in order to make the pseudo code much more readable. We hope that this can help in the better understanding of the algorithms used.
Round 2
Reviewer 1 Report
The authors appropriately revised the manuscript according to the reviewers’ comments.
Author Response
The authors appropriately revised the manuscript according to the reviewers’ comments.
Answer: Thank you for giving us the opportunity to improve our work with your precious suggestions.
Reviewer 2 Report
The paper's improvement is not significant.
In the previous revision suggestion, I mentioned the author need to demonstrate the importance of the application scenario. I didn't find the revision related to the issue. If it can be solved, it helps to improve the paper.
Also, I mentioned if the author can make the code open source (maybe upload it on Github), so the reviewer can run the code. The author seems didn't respond to the request. If reviewer can experience the HRI they created, it would be better evaluated. If the code needs to be private, authors need to clarify the reason.If more revision is done, I believe it could be better.
Author Response
In the previous revision suggestion, I mentioned the author need to demonstrate the importance of the application scenario. I didn't find the revision related to the issue. If it can be solved, it helps to improve the paper.
Answer: as mentioned in the paper, our goal is to design a computational model, based on consolidated theories, that allows a robot to personalize a museum visit. The main goal is personalization and the relevance of the experiment is related to the capability of the robot to provide a suitable museum tour with respect to the mental states attributed to the users and the constraints represented by the mental states attributable to other agents involved in the interaction. The museum tour personalization is not limited to suggest the best tour with respect to the artistic habits of the user, but also to provide a description of the tour suitable to the user needs such as interpreted by the museum’s curators. This kind of personalization is quite different from other approaches in the field of HRI and Cultural Heritage fields. Building user models and adapting the robot’s autonomy to these model is one of the main features to achieve a real and effective personalization.
Also, I mentioned if the author can make the code open source (maybe upload it on Github), so the reviewer can run the code. The author seems didn't respond to the request. If reviewer can experience the HRI they created, it would be better evaluated. If the code needs to be private, authors need to clarify the reason.
Answer: The code is constantly evolving and developing in collaboration with a Roman museum with which a project partnership is being born. For these reasons, we believe that the pseudo-code is sufficient, which is certainly more centered on the specific pilot study and clarifies the main heuristic mechanisms on which the application presented is based.